# Chemical Biopreservative Effects of Red Seaweed on the Shelf Life of Black Tiger Shrimp (*Penaeus monodon*)

**DOI:** 10.3390/foods9050634

**Published:** 2020-05-14

**Authors:** Abimannan Arulkumar, Kumar Satheeshkumar, Sadayan Paramasivam, Palanivel Rameshthangam, Jose M. Miranda

**Affiliations:** 1Department of Oceanography and Coastal Area Studies, School of Marine Sciences, Alagappa University, Science Campus, Karaikudi 630 003, Tamil Nadu, India; aruul3@gmail.com (A.A.); Satheshkumar.sk@gmail.com (K.S.); drparamsan@gmail.com (S.P.); 2Department of Biotechnology, Acahriya Arts and Science College (Affiliated to Pondicherry University), Puducherry 605 110, India; 3Department of Biotechnology (DDE), Science Campus, Alagappa University, Karaikudi 630 003, Tamil Nadu, India; rameshthangam@alagappauniversity.ac.in; 4Departamento de Química Analítica, Nutrición y Bromatología, Facultade de Veterinaria, Universidade de Santiago de Compostela, 27002 Lugo, Spain

**Keywords:** *Acanthophora muscoides*, biogenic amines, *Hypnea musciformis*, shelflife, shrimp

## Abstract

*Hypnea musciformis* (HM) and *Acanthophora muscoides* (AM) red seaweeds were evaluated for their antioxidant properties and efficacy to extend the chemical shelf life of black tiger shrimp (*Penaeus monodon*) during 14-daystorage. Treated shrimp were soaked in five percent ethanolic solution with 500 µg/mL of AM or HM powder for 30 min. HM had more phenols and flavonoids, increased radical scavenging activity, and greater H_2_O_2_ reducing power than AM in vitro. Biochemical quality indicators were significantly higher in the control group, followed by HM- and AM-treated samples during storage. On day 14 of storage, controls contained significantly higher amounts of biogenic amines than HM- or AM-treated samples. The shelf life of chilled stored shrimp increased due to the presence of compounds of butylated hydroxytoluene, sulfurous acid, heptadecane, mono (2-ethylhexyl), and 1,2-propanediol found in AM extract and sulfurous acid and 1,2-propanediol found in HM extract. A control group was soaked in the same ethanolic solution as treated samples without algae powder for 30 min. Each group was kept ice-cold during the soaking period. The results obtained demonstrate the usefulness of two seaweed extracts, *Hypnea musciformis* and *Acanthophora muscoides*, combined with ice by decreasing the formation of toxic biogenic amines in shrimp, enhancing its shelf life during ice storage.

## 1. Introduction

In the fishery sector, both manufacturers and consumers are becoming increasingly focused on minimally processed products, with the aim of reducing the use of synthetic food additives without compromising food safety [1]. A number of preservatives are currently used for the purpose of slowing the spoilage of fishery products during storage and extending their shelf-life [2]. The use of natural food additives, such as algae or plant extracts, provides unlimited opportunities for the inhibition of bacterial growth because of their chemical diversity. Previous works demonstrated antibacterial activity of both algae and extracts obtained from them against food-borne pathogens and spoilage bacteria [3]. It was also reported that plant source components can improve the organoleptic properties of fishery products, including shrimp [4].

Marine algae are an important constituent of the human diet in various Asiatic nations, providing important macro- and micronutrients [5,6]. In other countries, although algae consumption is less usual, it has gained prominence in recent decades [6]. Furthermore, marine algae (green, red, and brown) contain many natural bioactive compounds, such as phlorotannins, with important antioxidant and antibacterial properties for potential application in biological systems [6]. Several investigators have highlighted the potential biopreservative properties of algae on seafood, mainly aimed at extending the shelf-life of fishery products, but also reducing the development and presence of harmful substances such as biogenic amines [4,5]. The consumption of seafood stored in unsuitable conditions that allow the increase of biogenic amines is associated with harmful effects, including death, and represents a serious health risk to humans [7]. The formation of biogenic amines in fishery products requires a dual mechanism. Thus, it is necessary to determine the primary presence of amino acid precursors in fish products. Additionally, it is also necessary to determine the presence and multiplication of bacteria producing amino acid decarboxylase enzymes to decarboxylate aminoacids [7].

Black tiger shrimp (*Penaeus monodon*) is a commonly consumed crustacean species harvested in considerable quantities across the world. In 2014, its worldwide production, including aquaculture and extractive fishing, reached about one million Tm [8]. This crustacean species is highly popular but is also a highly perishable commodity, greatly susceptible to oxidation, which contributes to spoilage and loss of quality during cold storage [9]. Therefore, effective methods for controlling oxidation are required to improve the shelf-life of fresh black tiger shrimp.

Although red seaweed is a potential and effective source of bioactive compounds, it is currently not exploited enough for its natural preservative efficacy in shrimp. Therefore, this research work aimed to investigate the shelfl-ife enhancing potential of two types of red seaweed, *Hypnea musciformis* (HM) and *Acanthophora muscoides* (AM), on peeled black tiger shrimp during storage. We also investigated the antioxidant properties of HM and AM as natural bioprotective sources for use in the seafood processing industry.

## 2. Materials and Methods

### 2.1. Collection and Preparation of Samples

Fresh black tiger shrimp (*n* = 240) were obtained from the Thondi fish landing center (Southeast coast, Tamil Nadu, India), (latitude 9°44′ 16.83′′ N, longitude 79°01′ 05.47′′ E). Samples were immediately carried to the laboratory in refrigerated conditions and measured, with an average length of about 18 cm and an average weight of about 20 g.

HM and AM were collected from Thondi coastal water and intensively washed with potable water to eliminate debris and sand. The seaweeds were identified following the method described by Central Marine Fisheries Research Institute guidelines [10]. Samples were then carried to the laboratory and dried at room temperature under shade for 96 h to remove moisture content. Subsequently, dried samples were ground into fine powder (around 2 mm) using Tissue Blender, stored in a refrigerator (4 °C), and used within 1 month for further analysis. Then 500 µg/mL of HM or AM powder was dissolved with 5% ethanolic solution and made up to 100 mL. Similarly, a control group was prepared without algae powder. The samples were filtered through Whatman No. 1 filter paper and maintained in separate containers.

Black tiger shrimp were cleaned with tap water, peeled, eviscerated, and then split into 3 groups. One treatment group was soaked in 100 mL solution for 30 min, and a second treatment group was soaked in 100 mL solution for 30 min. Both algae concentrations and solvent were chosen based on previous works [11,12]. A control group was soaked in the same ethanolic solution as treated samples without algae powder for 30 min. Each group was kept ice-cold during the soaking period. After 30 min, all shrimp in the experimental groups were drained well, and then 240 shrimp (15–20 g) from each group were packed in a low-density polyethylene pouch, placed in polypropylene boxes (Milton), and stored on ice. Samples of about 17 g were withdrawn for biochemical analysis at 0, 4, 7, 11, and 14 days of storage, and all biochemical determinations were made in triplicate.

### 2.2. Determination of Total Phenol Content

Total phenolic content was measured in methanolic solutions of HM and AM powder according to the method published by Lim et al. [13]. First, 0.5 mL of each solution (5 mg/mL in methanol) was added to a vial containing 2.25 mL of methanol. Then 0.22 mL of Folin–Ciocalteu reagent was added, and the mixture was stirred for 1 min and maintained at room temperature for 8 min. A volume of 2.0 mL of sodium carbonate (Na_2_CO_3_; 7.5% *w*/*v*) was then added, and the subsequent mixture was heated at 25 °C for a minimum of 120 min. The optical density (OD) of the seaweed solutions and a methanol control sample were measured at 756 nm by means of an ultraviolet–visible (UV-VIS) 2450 spectrophotometer (Shimadzu, Tokyo, Japan). Total phenolic compound concentration was obtained by the formula T = C × V/M, in which T is total phenolic content (mg/g) (blank OD–sample OD) of the seaweed solution in gallic acid equivalent (GAE/g); C is the concentration of gallic acid obtained from the calibration curve (mg/mL); V is the volume of the solution (mL); and M is the weight of algae powder (g).

### 2.3. Determination of Total Flavonoid Content

The total flavonoid content of HM and AM solutions was measured based on a previously published method [14] and expressed as milligrams of (+) catechin equivalent (CE) per gram of seaweed dry weight (DW) (mg CE/g DW). A 1 mL aliquot of seaweed solution was added to a flask containing 4 mL of deionized water. Then, 0.3 mL of a sodium nitrite solution (5% NaNO_2_) was added and the flask was kept motionless for 5 min. Afterwards, 0.3 mL of 10% AgCl_3_ in water solution was added and the flask was kept motionless for 6 min, then 2 mL of sodium hydroxide solution (1 M NaOH) was added. The total volume was then completed to a total of 10 mL using deionized water. The final solution obtained was then vigorously shaken, and the absorbance was determined by comparison to a blank sample at 510 nm by using a UV-VIS spectrophotometer.

### 2.4. 1,1-Diphenyl-2-Picrylhydrazyl Radical Scavenging Activity

The determination of antioxidant activity of HM and AM solutions was performed usingdiphenyl-2-picrylhydrazyl (DPPH) as standard reagent (Sigma-Aldrich, St. Louis, MO, USA), according to the method reported by Lim et al. [13]. A methanol stock solution of each solution was prepared and dilutions were performed to obtain concentrations ranging from 0.1 to 1.0 mg/mL. Solutions obtained (2.0 mL) were mixed with 2.0 mL of 0.16 mM DPPH/methanol solution. The mixtures were then agitated and placed in the dark for 30 min. The absorbance of each mixture was determined at 517 nm by comparison to a reagent blank sample by means of a UV-VIS spectrophotometer. The final scavenging activity was then calculated by the formula DPPH radical scavenging activity (%) = Abs_517_ of control − (Abs_517_ of sample/Abs_517_ control) × 100, and the IC_50_ value (concentration of sample required to scavenge 50% of DPPH radicals; mg/mL) was calculated.

### 2.5. Hydrogen Peroxide (H_2_O_2_) Scavenging Activity

The capacity of HM and AM solutions to scavenge H_2_O_2_ was measured in accordance with the method reported by Ruch et al. [15]. For this purpose, 40 mM of H_2_O_2_ was diluted in a phosphate buffer, and pH was adjusted to 7.4. Seaweed solutions were prepared at dilutions in the range 0.1–1.0 mg/mL, in which a 1 mL aliquot was added to 3.0 mL of 40 mM H_2_O_2_ solution. After 10 min of incubation, absorbance was determined at 230 nm compared to a control sample consisting only of phosphate buffer (without H_2_O_2_). The rate of H_2_O_2_ scavenging was then determined using the formula [(A_0_− A_1_)/A_0_] × 100, where A_0_and A_1_are the absorbance obtained for the control and target sample, respectively. After determining the H_2_O_2_ scavenging activity, the IC_50_ value was calculated, measured in mg/mL.

### 2.6. Determination of Ferrous Ion (Fe^2^) Reducing Power

The reducing power of HM and AM solutions was obtained based on the capacity of antioxidants to produce color by means of a complex reaction with potassium ferric cyanide, trichloroacetic acid (TCA), and ferric chloride, based on a previously described method with minor modifications [13]. A 1 mL aliquot of HM and AM solution (1 mg/mL in methanol) was shaken with 2.5 mL of phosphate buffer (pH 6.6) and 2.5 mL of 1% potassium ferric cyanide. After that, the blend obtained was maintained at 50 °C for 20 min. Then, a 2.5 mL aliquot of the upper layer was taken and mixed with 2.5 mL of deionized water and 0.5 mL of 0.1% ferric chloride. The reducing power was determined after a 10 min incubation step, with the absorbance measured at 700 nm.

### 2.7. Chromatographic Characterization of Antioxidant Compounds

Separation of bioactive compounds contained in HM and AM solutions was performed by thin layer chromatography (TLC), according to a method published by Osman et al. [16]. The mobile phase was obtained by a mixture of methanol and chloroform (1:9). About 1 mg/mL of target sample was spotted on the TLC plates and air-dried. The spots obtained were identified using both long- and shortwave UV light, and in an iodine chamber. The distance moved by the solute relative to the distance moved by the solvent was considered as the Rƒ value, determined to find the active metabolites. The developed chromatograms were subjected to antimicrobial activity test by the bioautography method [17]. The standardized chromatography plates of HM and AM crude powder were sprayed with DPPH (dissolved in dimethyl sulfoxide (DMSO). The specific target compound (band) that possessed antioxidative activity showed a clear zone of inhibition.

The crude compounds were partially purified using TLC plates (Merck, Darmstadt, Germany) through a solvent system composed of a chloroform/ethyl acetate/methanol mixture in proportions of 6:3:1. Afterwards, the partially purified fractions were loaded into a silica gel packed column (20 cm length and 2 cm diameter) and eluted with a mixture of *n*-hexane and ethyl acetate (50:50 *v*/*v*). The fractions that were obtained and purified were identified by means of a gas chromatograph (GC-2010) coupled to a quadrupole mass spectrometer (QP-2010, Shimadzu, Kyoto, Japan) and an Rtx-PCB capillary column (60 m × 0.25 mm i.d., 0.25 mm film thickness; Restek, Bellefonte, PA, USA). As gas carrier, helium with purity higher than 99% was used at a flow rate of 1 mL/min. Samples of 1 mL were injected in split mode. The temperatures were slightly different for the injector port (250 °C), interface (270 °C), and ion source (230 °C). GC temperature began at 50 °C (1 min) and was increased by 10 °C min^−1^ until reaching a final temperature of 320 °C. The mass spectrometer was used in electron ionization mode at 70 eV and emission current of 60 mA. Full scan data were obtained in a mass range of 50–500 *m*/*z*. Interpretation of the obtained Gas Chromatography-Mass Spectrometry (GC-MS) analysis was carried out compared with data provided by the National Institute of Standards and Technology (NIST) library database [18].

### 2.8. Determination of Total Volatile Base Nitrogen

The content of total volatile base nitrogen (TVB-N) in muscle samples was determined according to Conway’s dish method [19]. First, 10-g samples of shrimp muscle were combined with6% trichloroacetic acid (TCA) and filtered through filter paper to extract TVB-N from each sample. In order to ensure complete and efficient extraction, residue was re-extracted twice. Afterward, the TCA extract of the shrimp muscle samples was absorbed by boric acid and then titrated with 0.02 N hydrochloric acid. The TVB-N content was then determined and expressed as mg/100 g.

### 2.9. Determination of Trimethylamine Concentration

The trimethylamine (TMA-N) content of the shrimp muscle samples was obtained by using the Conway technique [19]. According to this method, 1 mL of 10% potassium carbonate was added to shrimp samples and neutralized formalin was pipetted into the extract, with the objective of generating an interaction with the ammonia in the shrimp, thereby allowing only the TMA-N to diffuse over the Conway dish diffusion unit. TMA-N concentration was then determined and expressed as mg/100 g.

### 2.10. Determination of Sensory Score

Sensory score was determined using the quality score index (QI) method according to the guidelines provided in [20]. For this purpose, 3 shrimp were taken from each batch and each sampling day and evaluated by 5 experts. The head, body, tail, and meat color and meat appearance, texture, and odor were evaluated in raw shrimp. Each parameter had a score of 0–3, with 0 representing best quality and 3 poor qualities. Shrimp were considered acceptable at QI ≤ 14. The 5 experts were asked to conduct 3 determinations each, for a total of 15 determinations for shrimp batches. The shelf life of shrimp was calculated by a regression equation: *y* = 1.7139*x* + 0.5546 [20].

### 2.11. Determination of Biogenic Amine Content

Biogenic amine content was determined using shrimp muscle samples of about 5 g, which were combined in centrifuge tubes with a solution 20 mL of 6% TCA. Tubes were subsequently shaken for 3 min. Afterwards, the homogenates obtained were centrifuged at 10,000 × *g* at 4 °C for 10 min in a Remi CPR-30 Plus centrifuge (REMI, Mumbai, India), and filtered through Whatman filter paper No.1. The filtered extracts were then placed into volumetric flasks and TCA was added to achieve a total volume of 50 mL. A 1-mL aliquot of shrimp extract was derivatized with dansyl chloride (Sigma-Aldrich, USA), based on a method reported by Ereola et al. [21]. Briefly, to 1 mL of each free base amine solution (1 mg/mL), 0.2 mL of 2 M NaOH and 0.3 mL saturated sodium bicarbonate were added, followed by 2 mL of 1% dansyl chloride dissolved in acetone. Afterwards, the mixture was vigorously shaken for 1 min, then maintained at 40 °C for 45 min. Next, 100 µL of ammonia was added to the mixture and it was maintained at room temperature for 30 min. Total volume was then completed to 5 mL by adding acetonitrile. The solution was then centrifuged again under the conditions described above and the resulting supernatant was filtered through a 0.45-µm filter before HPLC analysis.

The content of biogenic amines in shrimp muscle samples was determined using a liquid chromatograph (Hitachi, Tokyo, Japan) equipped with a pump (L-7100, Hitachi), a Rheodyne syringe loading sample injector (model 7125), a UV-VIS detector set at 254 nm (L-4000, Hitachi), a chromatointegrater (D-2500, Hitachi), and a LiChrospher 100 RP-18 reverse phase column (125 × 4.6 mm i.d., 5-µm particles; Merck, Darmstadt, Germany) for chromatographic separation. The gradient elution program increased from an initial 50:50 (*v*/*v*) acetonitrile/HPLC grade water at a flow rate of 1.0 mL/min for 19 min, followed by a linear increase to 90:10 acetonitrile/water at 1.0 mL/min for 1 min, before the acetonitrile/water was gradually changed to 50:50 at 1.0 mL/min for 10 min. The total separation lasted 30 min. The gradient was run for 25 min at 22.5 °C to achieve adequate separation. The total injection volume was 5 µL for the standard and 20 µL for the test samples.

### 2.12. Statistical Analysis

The results obtained were compared using a one-way analysis of variance (ANOVA), using treatment as the independent variable. In all cases determinations were performed in triplicate, and differences obtained were considered significant at *p* < 0.05, determined by means of Duncan’s multiple range test using SPSS version 14 (SPSS Institute, Chicago, IL, USA).

## 3. Results

The total phenolic content of HM and AM methanolic solutions (Table 1) was 10.05 ± 0.19 and 7.12 ± 0.01 mg GAE/g, respectively. So, total phenolic content was significantly higher in HM than AM solution. A previous work found total phenol content in HM methanolic solution of 9.84 mg GAE/g, which is near that observed in the present work [5]. Other researchers have also documented the phenolic content in seaweed solutions as ranging from 37.66–178.75 mg GAE/g [22]. Higher phenolic and polyphenol content of seaweed indicates better scavenging capacity, which contributes to preventing lipid oxidation of foods.

Flavonoids are considered the most important natural phenolic component, owing to their demonstrated extensive chemical and biological activities. The total flavonoid content of the red seaweed solutions was higher for HM (9.5 ± 0.24 mg CE/g) than AM (8.21 ± 0.29 mg CE/g) (*p* < 0.05). In both cases, flavonoid content was higher than the levels previously obtained by Cox et al. [23], who found 6.83–7.41 mg CE/g in other seaweed species, such as *Palmaria palmate* and *Chondrus crispus*.

Methanolic solutions of HM and AM showed DPPH scavenging activity of 13.43 ± 0.38 and 7.12 ± 0.01%, respectively, lower than the values obtained by other authors for red algae, such as 15.4% for *Hypnea musciformis* [5] and 25% for *Palmaria palmate* [23]. The observed differences between HM and AM solutions could be attributed to a higher amount of phenolic and polyphenolic compounds in HM than AM. The results obtained in the present work suggest that methanolic solutions of these two types of red seaweed may contain phenolic and polyphenolic compounds with multiple –OH groups, which have the ability to donate a proton to DPPH, and thus neutralize free radicals.

Red seaweeds are known as a rich source of natural antioxidant compounds that are capable of scavenging H_2_O_2_ [5]. The H_2_O_2_ radical scavenging activity of HM and AM (at 1 mg/mL) was 40.45 ± 0.54% and 23.17 ± 0.40%, respectively, significantly higher for the HM solution.

Reducing agents present in the seaweed solution reduce the ferric cyanide complex to the ferrous form by donating a hydrogen [22]. Increased absorbance at 700 nm indicates an increase in reducing ability. At 0.1 mg/mL, the HM solution showed significantly higher reducing power (absorbance) than the AM solution (0.68 ± 0.01% *vs* 0.44 ± 0.02%, respectively; Table 1). Chakraborty et al. [5] also reported a high total reducing power of HM at 1 mg/mL of 0.74%, slightly higher than the values reported in the present work.

The crude methanolic solutions of HM and AM at a concentration of 1 mg/mL showed the presence of three major compounds, with R*f* values of 0.65, 0.76, and 0.84, visualized under the iodine chamber and UV light. The presence of compounds with antioxidant activity was noted by yellow spots against a purple background on TLC plates sprayed with 0.2% DPPH. After developed TLC plates were carefully dried for complete removal of the solvents, the areas of inhibition were compared with R*f* values of the related spots on the reference TLC plate. The purified compounds showed the inhibition activity, which had an R*f* value of 0.76 against the free radical as assessed by DPPH.

TVB-N is the most commonly employed chemical parameter to assess the quality and shelf-life of fishery foods [11]. Even a slight increase in TVB-N could reflect amine formation by microbes and/or autolysis in fish. Based on TVB-N and TMA-N content, previous works reported that *Litopneanus vannamei* stored in ice has a shelf life of at least 8 days [24], although other works did not confirm this result for *P. monodon* [25]. The TVB-N content of shrimp muscle increased significantly from an initial 10.96 to 34.43 mg/100 g for controls, which was significantly higher than the 24.00 and 23.96 mg/100 g detected in the HM- and AM-treated samples, respectively, at 14 days of storage (Figure 1). In only one sampling point after day 0 (day 11), controls did not show significant higher TVB-N values than HM- or AM-treated samples, but this single result did not affect the significantly higher increase in TVB-N values (*p* < 0.05) in control shrimp. The maximum allowable TVB-N limit varies between 30 and 50 mg/100 g in fishery foods [26,27]. In this study, the TVB-N levels exhibited an increasing trend, and the maximum allowable limit of TVB-N was detected only in the control samples. Additionally, as an improvement in the results obtained with respect to controls, both HM- and AM-treated samples showed better results in terms of shelf life based on TVB-N than those reported by Tam et al. [20], who showed that *P. monodon* stored in ice was not acceptable after an 8-day period.

The TMA-N level increased significantly in the control shrimp during storage compared to the HM- and AM-treated samples (*p* < 0.05), as shown in Figure 2. TMA-N reached 13.9 mg/100 g in the controls, and 11.93 and 11.86 mg/100 g in the HM- and AM-treated shrimp, respectively. Stronger prevention of TMA-N increase was reported during cold storage of megrim [28] treated with and without *F. spiralis*, where the final TMA-N ranged from 193.3–240.7 mg/100 g for the controls, and 0.67–2.50 g for the shrimp treated with *F. spiralis*. However, in the present study, TMA-N mainly indicated that the red seaweed solutions, particularly HM, effectively inhibited the decarboxylation of trimethylamine oxide to trimethylamine.

Figure 3 compares the evaluation in overall sensory score of *P. monodon* samples treated with HM or AM solution compared to control samples during ice storage. The QI sensory score increase was rapid in control samples, from an initial score of 0.96 to 18.45 on day 14. The HM- and AM-treated shrimp samples had better sensory scores and exhibited better color, texture, and odor during storage relative to the control, with scores of 14.18 and 14.95, respectively, on day 14. Thus, the controls as well as HM- and AM-treated shrimp were considered adequate for human consumption up to 9, 13, and 12 days, respectively. This period is longer than that reported by Tam et al. [20], who reported an average shelf life sensory score of 12 at 8 days of ice storage.

Biogenic amines are alkaline organic compounds mainly produced by decarboxylation reactions of free amino acids present in the muscle by the enzyme decarboxylase [20]. When seafood contains high levels of bacteria with the ability to produce decarboxylase enzymes, these free amino acids undergo decarboxylation reaction to finally produce biogenic amines [28]. In the present work, lower concentration of biogenic amine content was found than that recently reported by other authors for fresh white prawn (*Exopalaemon modestus*) [29]. Changes in biogenic amine content (Table 2) revealed that the addition of HM and AM treated samples prevented the formation of biogenic amines in shrimp with respect to controls up to day 11 of storage. Between the assayed biogenic amines, tryptamine was found in all tested samples on all sampled days of storage (except day 0; ranging from 1.29–9.13 mg/100 g) including the control samples. On day 14, all tested biogenic amines were found in control samples.

Spermidine and spermine naturally occur in foods, and neither substance is related to bacterial spoilage. In the current work, the concentration of spermidine and spermine in fresh meat was less than 1 mg/100 g of muscle. Spermidine was first detected in the control samples on day 7 of storage (1.27 mg/100 g), and increased by day 14.

Histamine is mainly produced from bacterial decarboxylation of histidine in fish and shellfish tissue [20] related to bacterial metabolism such as *Achromobacter histamineum* and *Proteus morganii*, among others [20]. The significance of determining the content of histamine in aquatic products is the high potential impact on human health, because histamine poisoning can cause death [7]. In the current work, histamine could be detected only in the control samples after day 11 of storage. After finishing the storage period, histamine content in controls reached 4.52 mg/100 g, close to the limit of 5 mg/100 g that is generally accepted as possibly causing histamine poisoning in humans [7]. All of the results obtained are much lower than those obtained for *P. monodon* by Tam et al. [20], who, after a 10-day period of ice storage, found a maximum content of 3227 mg/100 g.

Other biogenic amines that can act as potentiators for histamine food poisoning are tyramine, tryptamine, putrescine, and cadaverine [30]. In this work, tryptamine was not detected in shrimp between days 0 and 11 of storage; however, tryptamine was found at end of the storage, at 1.75 mg/100 g in the control samples. Tyramine content was not detected in either the HM- or AM-treated samples throughout the storage period (Table 2). This suggests that treatment with HM and AM solution inhibited the formation of tyramine in shrimp during ice storage.

Through Gas Chromatography-Mass Spectrometry (GC-MS) determinations, 13 and 3 compounds were identified from HM and AM, respectively (Table 3 and Table 4).

Among them important natural preservative agents were found: butylated hydroxytoluene, sulfurous acid, heptadecane, mono (2-ethylhexyl), phthalate 1, 2-propanediol, octacosane, 2,4,6, tris (1,1-dimethyethyl)-4-methylcyclohexa-2,5-dien-1-one, and 1,2-propanediol (Table 3 and Table 4). These results are in agreement with previously reported results for brown algae from the Persian Gulf [31]. The high levels of tetradecanoic acid, 9-z-octadecanic acid, pentadecanoic acid, hexadecanoic acid, and octadecanoic acid in red seaweed solution makes it potentiallyan antioxidant and antimicrobial. Mohy et al. [32] identified in other seaweeds, *Jania rubens* and *Pterochadia capillacea*, various bioactive compounds, such as ascorbic acid 2,6-dihexadecanoate, icosapent, *trans*-13-octadecenoic acid, 3,7,11,15-tetramethyl-2-hexadecen-1-ol, heptadecane, and 1,2-benzenedicarboxylic acid.

## 4. Conclusions

The in vitro antioxidant studies revealed that HM ethanolic solution possessed higher phenolic and flavonoid content than the AM solution. In addition, both HM and AM solutions improved the chemical shelf-life of tiger shrimp during ice storage (extended by four and three days, respectively) compared to control samples. Seaweed solution acted as a biopreservative to maintain the shrimp quality and reduce the formation of harmful biogenic amines and TVB-N compounds, although more information is needed about the safety and toxicological effects of both seaweeds. This work provides a basis for the use of red seaweeds as natural preservatives to extend the shelf-life of ice-chilled seafood for transportation and in the retail market.

## Figures and Tables

**Figure 1 foods-09-00634-f001:**
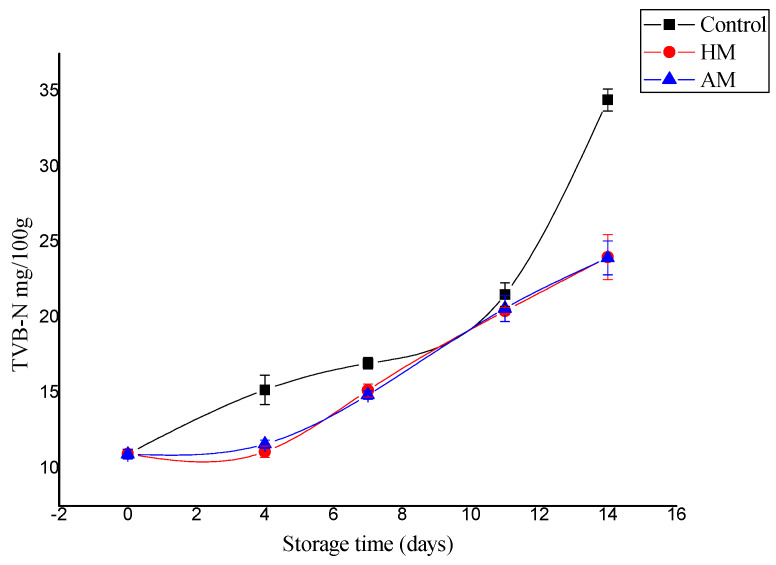
Changes in total volatile base nitrogen (TVB-N) values of control, *Hypnea musciformis* (HM), and *Acanthophora muscoides* (AM)treated shrimp muscle relative to days of storage at 0 °C (average ± standard deviation).

**Figure 2 foods-09-00634-f002:**
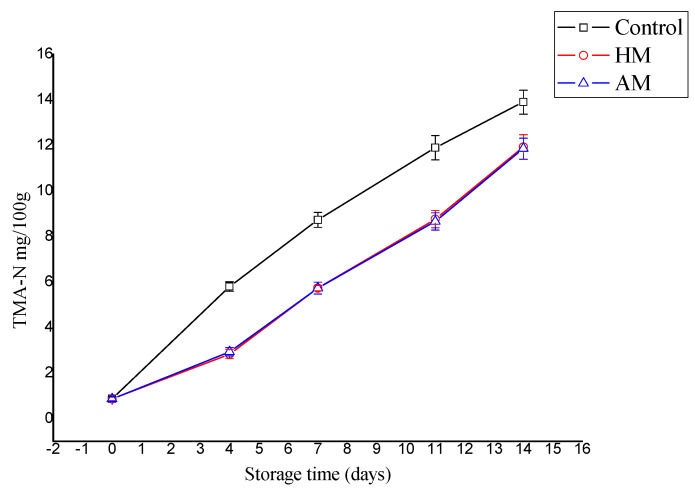
Evaluation of trimethylamine (TMA-N) content in control, *Hypnea musciformis* (HM), and *Acanthophora muscoides* (AM) treated shrimp muscle relative to days of storage at 0 °C (*n* = 3, average ± standard deviation).

**Figure 3 foods-09-00634-f003:**
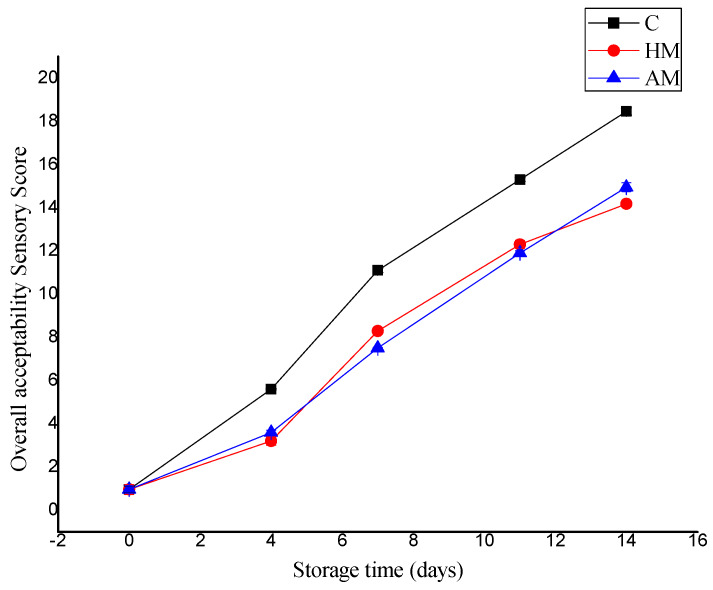
Evaluation of quality score index of control, *Hypnea musciformis* (HM), and *Acanthophora muscoides* (AM) treated shrimp muscle relative to days of storage at 0 °C (*n* = 3, average ± standard deviation).

**Table 1 foods-09-00634-t001:** Total phenolic content and antioxidant activity of *Hypnea musciformis* and *Acanthophora muscoides*.

Parameter (Unit) *	*H. musciformis*	*A. muscoides*
Total phenolic content (mg GAE/g)	10.05 ^a^ ± 0.19	7.12 ^b^ ± 0.01
Total flavonoid content (mg CE/g)	9.5 ^a^ ± 0.24	8.21 ^b^ ± 0.29
DPPH scavenging ability (%)	13.43 ^a^ ± 0.38	7.12 ^b^ ± 0.01
H_2_O_2_ scavenging ability (%)	40.45 ^a^± 0.54	23.17 ^b^ ± 0.40
Ferric ion reducing power (%)	0.68 ^a^ ± 0.01	0.44 ^b^ ± 0.02

* Values presented are averages of three measurements ± standard deviation. Superscript letters denote significant differences (*p* < 0.005).

**Table 2 foods-09-00634-t002:** Biogenic amine content (mg/100 g muscle) of *Penaeus monodon* treated with *Hypnea musciformis* (HM) and *Acanthopora muscoides* (AM) during storage at 0 °C for 14 days.

Group	Days	Try	Put	Cat	His	Tyr	Spd	Spn
Control	0	ND	ND	ND	ND	ND	ND	ND
Control	4	1.61 ± 0.11 ^b^	ND ^b^	ND	ND	ND	ND	ND
HM		2.89 ± 0.24 ^a^	ND ^b^	ND	ND	ND	ND	ND
AM		1.86 ± 0.12 ^b^	ND ^b^	ND	ND	ND	ND	ND
Control	7	2.52 ± 0.01 ^a^	1.84 ± 0.12 ^a^	ND	ND	ND	1.27 ± 0.11 ^a^	ND
HM		2.70 ± 0.07 ^a^	1.41±0.01 ^b^	ND	ND	ND	ND ^b^	ND
AM		1.32 ± 0.21 ^b^	ND ^c^	ND	ND	ND	ND ^b^	ND
Control	11	9.13 ± 0.11 ^a^	ND ^b^	ND	1.31 ± 0.09 ^a^	ND	3.49 ± 0.17 ^a^	1.57 ± 0.05 ^a^
HM		1.29 ± 0.03 ^b^	2.10 ± 0.60 ^a^	ND	ND ^b^	ND	ND ^b^	ND ^b^
AM		1.51 ± 0.10 ^b^	ND ^b^	ND	ND ^b^	ND	ND ^b^	ND ^b^
Control	14	11.81 ± 0.41 ^a^	3.77 ± 0.14 ^a^	2.02 ± 0.12 ^a^	4.52 ± 0.10 ^a^	1.75 ± 0.06 ^a^	8.5 ± 0.13 ^a^	3.87 ± 0.06 ^a^
HM		1.97 ± 0.51 ^b^	ND ^b^	ND ^b^	ND ^b^	ND ^b^	ND ^b^	ND ^b^
AM		1.16 ± 0.09 ^b^	ND ^b^	ND ^b^	ND ^b^	ND ^b^	ND ^b^	ND ^b^

Values presented are averages of three replicates ± standard deviation. Superscript letters denote significant differences (*p* < 0.05). C, control group (not treated with red seaweed extract); HM, shrimp muscle treated with *Hypnea musciformis*; AM, shrimp treated with *Acanthophora muscoides*; Try, tryptamine; Put, putrescine; Cad, cadaverine; His, histamine; Tyr, tyramine; Spd, spermidine; Spn, spermine; ND, not detected.

**Table 3 foods-09-00634-t003:** Active preservative compounds from *Hypnea musciformis* (HM) identified using GC-MS analysis.

Number	Compound	Rt	Base *m*/*z*
1	Butylated hydroxytoluene	8.841	205.05
2	Sulfurous acid	12.217	97.05
3	Heptadecane	18.706	57.00
4	Hexatriacontane	20.185	57.00
5	Tetratriacontane	21.608	57.00
6	Tetracosane	22.978	57.00
7	1,2-Propanediol	23.494	91.00
8	Octacosane	24.293	57.00
9	Mono (2-ethylhexyl) phthalate	24.861	148.95
10	Pentatriacontane	25.561	57.00
11	Octacosane	26.789	57.00
12	Triacontane	27.951	57.00
13	Nonacosane	29.048	57.00

GC-MS: Gas Chromatography-Mass Spectrometry; Rt: Retention time; *m*/*z*: Mass/charge ratio.

**Table 4 foods-09-00634-t004:** Active preservative compounds from *Acanthopora muscoides* (AM) identified using GC-MS analysis.

Number	Compound	Rt	Base m/z
1	2,4,6, Tris (1,1-dimethyethyl)-4-methylcyclohexa-2,5-dien-1-one)	8.830	205.05
2	Sulfurous acid	12.208	96.95
3	1,2-Propanediol	23.511	91.00

GC-MS: Gas Chromatography-Mass Spectrometry; Rt: Retention time; *m*/*z*: Mass/charge ratio.

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
