# Peer review of "Chemical Biopreservative Effects of Red Seaweed on the Shelf Life of Black Tiger Shrimp (Penaeus monodon)"

_foods, 2020, doi:10.3390/foods9050634_

Round 1

Reviewer 1 Report

This manuscript provides a very complete study, including accurate and complementary analytical tools to prove the positive effect of two algae. I think it can be accepted for publication provided some minor aspects are performed.

Abstract

It is not expressed haw the red algae or their extracts were added to the shrimp.

Line 24: Clarify where is the presence of such compounds found

Lin 26: freeze is mentioned. Is not it fresh products ?

Experimental part

Clarify if the algae were previously extracted or the shrimp just dipped in a mixture of alga dust and 5 % ethanol solution. Provide more information on how the soaking media were prepared. In fact, in Conclusions, HM and AM solutions are mentioned.

Results and discussion

Line 273: For a crustacean, TVB-N limits are a bit higher than for fish. Please, take this into account and include it.

Figures

In X axis, replace “time” by “storage time”.

Author Response

RESPECTED REVIEWER 1

With respect to comments about “This manuscript provides a very complete study, including accurate and complementary analytical tools to prove the positive effect of two algae. I think it can be accepted for publication provided some minor aspects are performed.”

Response: Many thanks for your comments. We have incorporated all the comments suggested by Reviewer.

With respect to comments about “it is not expressed how the red algae or their extracts were added to the shrimp.”

Response: In the revised version of the manuscript is was clarified thatAcanthophora muscoides (AM) and Hypnea musciforms (HM) powder each 500 µg/ml was dissolved with 5% ethanolic solution and made up to 100 ml. Similarly a control (c) group was prepared without algal powder. Further the samples were filtered with Whatman No 1 filter paper and collected in separate containers.”

With respect to the addition to shrimps, it was stated “Treated shrimps were soaked in 500 µg/ml of AM or HM powder in 5% ethanolic solution for 30 min.”

With respect to comments about “Line 24: Clarify where is the presence of such compounds found”

Response: According to the suggestion from the Reviewer, it was clarified that “The shelf-life of chilled stored shrimp increased due to presence of compounds of butylated hydroxytoluene, sulfurous acid, heptadecane, mono (2-ethylhexyl) and 1,2-propanediol Found in AM extracts and sulfurous acid and 1,2-propanediol found in HM”

With respect to comments about “Line 26: freeze is mentioned. Is not it fresh products?”

Response: In fact, the Reviewer is right and in the revised version of the manuscript “freeze” was changed to “ice”. 

With respect to comments about “Clarify if the algae were previously extracted or the shrimp just dipped in a mixture of alga dust and 5 % ethanol solution. Provide more information on how the soaking media were prepared. In fact, in Conclusions, HM and AM solutions are mentioned.”

Response: According to the comments from the Reviewer in the revised version of the manuscript It was added “AM and HM powder each 500 µg/ml was dissolved with 5% ethanolic solution and made up to 1000 ml. Similarly, a control group was prepared without algal powder. Further the samples were filtered with Whatman No 1 filter paper and collected in separate containers.”

And it was corrected the following paragraph:

“Black tiger shrimps were cleaned with tap water, peeled, eviscerated, and then divided into three groups. First treatment group (AM) was soaked in 1000 ml solution for 30 min. A second treatment group (HM) was soaked in 1000 ml solution for 30 min. Both algae concentrations and solvent were chosen based in previous works [10,11]. A control (c) group was soaked in 1000 ml (5 %) ethanolic solution for 30 min. Each group was kept ice-cold during the soaking periods. After 30 min, all shrimp in the experimental groups were drained well, after which, 240 shrimp from each group were packed in a low-density polyethylene pouch, subsequently introduced in polypropylene boxes (Milton), and stored in ice. Samples about 17 g were withdrawn for biochemical analyses at 0, 4, 7, 11 and 14 days storage, and all biochemical determinations were made in triplicates.”

With respect to comments about “Line 273: For a crustacean, TVB-N limits are a bit higher than for fish. Please, take this into account and include it.”

Response: TVB-N maximum acceptable limit was revised following your suggestions to 50 mg/100 g. Additionally, a reference also included:

  1. Mendes, R.; Huidobro, A.; Lopez-Caballero, E. Indole levels in deepwater pink shrimp (Parapenaeus longirostris) from the Portuguese coast. Effects of temperature abuse. Eur. Food Res. Technol. 2002, 214, 125–130.

With respect to comments about “Figures: In X axis, replace “time” by “storage time”.

Response: According to the suggestions from the Reviewer, “Storage day” was replaced by “Storage time” in all figures.

Reviewer 2 Report

The paper should be revised for further improvement. Below the comments.

Lines 18, 22, 35, and 63: delete “on ice”

Line 25: "showed" instead of "showing"

Line 49: "associated" instead of "related"

Line 51: "amino acid precursors" instead of "amino acids precursors"

Line 51-53: reformulate the sentence "Firstly, … enzymes"

Line 72: "water and were intensively washed" instead of "waters and were intensive washed"

Line 76: describe how grinding was performed? The particle size obtained? The moisture content after drying?

Line 85-86: specify the size (g) of the samples

Line 95-99: Please clarify the equation used to determine the TPC in shrimp samples. We don’t see the OD value in this equation !

Line 105: the term “allowed” is not appropriate. Use for example “kept motionless”

Line 136: “after water” should be changed by another term.

Lines 141-157: How the antioxidant molecules are identified after revelation using DPPH?

Lines 175-179: there are countless grammatical errors and sentences requiring reformulation. Please revise.

Line 200: Section 2.12 can be merged with section 2.7.

Line 223-225: delete this paragraph

Lines 297-298: reformulate the sentence “This period … 8 days”

More general comments:

  • Why using ethanol to solubilize the algae powders. It may modify the taste of the shrimp since the incubation time is relatively long (30 min).
  • Why measuring the antioxidant activity using different methods?
  • Please provide the TLC plate photos.
  • At 11 days storage, the TVB-N mg/100g is the same between control and the algae treated samples. How the authors explain the huge difference observed in day 14?
  • According to figure 3, the control is more appreciated by the consumers. How the authors can defend the use of algae extracts for shrimp preservation?

Author Response

RESPECTED REVIEWER 2

With respect to the comments about “Lines 18, 22, 35, and 63: delete “on ice””

Response: According to the suggestions from the Reviewer, “on ice” is deleted.

With respect to the comments about “Line 25: "showed" instead of "showing"”

Response: According to the suggestions from the Reviewer, “showing” was changed to “showed”

With respect to the comments about Line 49: "associated" instead of "related"

Response: Text has been changed accordingly the comments from the Reviewer.

With respect to the comments about Line 51: "amino acid precursors" instead of "amino acids precursors"

Response: Text has been changed accordingly the comments from the Reviewer.

With respect to the comments about Line 51-53: reformulate the sentence "Firstly, … enzymes":

Response: Text has been changed accordingly the comments from the Reviewer.

With respect to the comments about Line 72: "water and were intensively washed" instead of "waters and were intensive washed"

Response: Text has been changed accordingly the comments from the Reviewer.

With respect to the comments about “Line 76: describe how grinding was performed? The particle size obtained? The moisture content after drying?”

Response: In order to clarify it, in the revised version of the manuscript it was added the following information: “Samples were carried to the laboratory and dried at room temperature under shade for 96 h to remove moisture content. Shade dried samples were grounded into fine powder using tissue blender. The particle size of the powdered HM and AM were around 2 mm. The powdered samples were then stored in refrigerator (4oC) and used within one month for further analysis.”

With respect to the comments about “Line 85-86: specify the size (g) of the samples”

Response: According to the suggestions from the Reviewer, it was specified that samples size was about 17 g each.

With respect to the comments about “Line 95-99: Please clarify the equation used to determine the TPC in shrimp samples.”

Response: According to the suggestion from the Reviewer, it was specified thati n calculation, T was the total phenolic content (mg/g) measured by (Blank OD-sample OD).

With respect to the comments about “Line 105: the term “allowed” is not appropriate. Use for example “kept motionless”

Response: According to the suggestion from the Reviewer, the term “allowed” was changed to “kept motionless”.

With respect to the comments about “Line 136: “after water” should be changed by another term.”

Response: According to the suggestion from the Reviewer, Answer: “after water” was changed tow “After that”.

With respect to the comments aboutLines 141-157: How the antioxidant molecules are identified after revelation using DPPH?”

Response: The antioxidant molecules are identified through TLC and GCMS (please see Table 3 and Table 4).

With respect to the comments about “Lines 175-179: there are countless grammatical errors and sentences requiring reformulation. Please revise.”

Response: According to the suggestions from the Reviewer, the text was checked and corrected accordingly.

With respect to the comments about “Line 200: Section 2.12 can be merged with section 2.7.”

Response: According to the suggestions from the Reviewer, the section 2.7 has been merged with section 2.12

With respect to the comments about “Line 223-225: delete this paragraph”

Response: According to the suggestions from the Reviewer, lines 235-237 were deleted from the revised version of the manuscript.

With respect to the comments about “Lines 297-298: reformulate the sentence “This period 8 days””

Response: According to the suggestions from the Reviewer, lines 297-298 were rephrased.

More general comments:

With respect to the comments about “Why using ethanol to solubilize the algae powders. It may modify the taste of the shrimp since the incubation time is relatively long (30 min).”

Response: Thank you for your comment. Aqueous ethanolic 5% solution was used to get a maximum yield of extraction from A. muscoides and H. musciformis. Moreover, ethanol solutions were mixed with ice and get diluted and their smell also will be subsided. It should be taken into account that after adding ice or ice with the ethanolic extracts, no decreases in the microbiological parameters were found (controls and treated samples showed similar bacterial counts).

Several works were previously performed that employed similar methodology for the determination of the microbiological shelf-life of fishery products. In example:

Barros-Velazquez, J., Miranda, J.M., Ezquerra-Brauer, J.M., Aubourg, S.P. (2016). Impact of icing systems with aqueous, ethanolic and ethenolic-aqueous extracts of alga Fucus spiralis on microbial and biochemical quality of chilled hake (Merluccius merluccis). International Journal of Food Science and Technology, 51, 2081-2089.

Miranda, J.M., Ortiz, J., Barros-Velazquez, J., Aubourg, S.P. (2016). Quality enhancement of chilled fish by including alga Bifurcaria bifurcata in the icing medium. Food and Bioprocess Technology, 9, 387-395.

With respect to the comments about “Why measuring the antioxidant activity using different methods?”

Response: Thank you for your comment. Different methods are used in marine algae extracts in order to find out the most effective method for antioxidant activity.

With respect to the comments about “Please provide the TLC plate photos.”

Response: You can find an image of the TLC plates carried out during the present experimental work

As you can seem, as you can see, the quality of the TLC plate image is not as good as we would like, and for that reason we preferred not to include it in the printed version of the manuscript.

With respect to the comments about “At 11 days storage, the TVB-N mg/100g is the same between control and the algae treated samples. How the authors explain the huge difference observed in day 14?”

Response: Thank you for your comment. The TVB-N content has differences in control, AM- and HM- treated samples from day 0 to 10. But similar values between controls, AM- and HM- were observed only on day 1, and please note that the line graph. On 14 day storage there was a notable difference observed between control and AM and HM. Please note that the graphed line in control samples values between days is not straight, but describes a descending parabola between days 7 and 11. In any case, this single results did not affected the statistical comparison, that showed higher TVB-N content in control samples than in treated samples. In the revised version of the manuscript, is was added the following phrase to clarify it:

“Only in one sampling point after day 0 (day 11), controls did not shown significant higher TVB-N values than HM- or AM-treated samples, but this single results did not affect to the significantly higher increase significantly in TVB-N values (P < 0.05) in control shrimps compared to the HM- and AM-treated samples.”

With respect to the comments about According to figure 3, the control is more appreciated by the consumers. How the authors can defend the use of algae extracts for shrimp preservation?

Thank you for your comment. Based on Figure 3, sensory score of control sample have higher Quality Score Index (QI) than AM and HM treated samples. Please note that QI method is a inverse score, in which “For each parameter, a score of 0-3 were stated, being 0 the best quality and 3 the poor quality. Shrimps were considered acceptable with QI ≤ 14.”. Consequently, a higher QI value denotes a poor sensory quality.

Round 2

Reviewer 1 Report

The manuscript has been performed according to previous suggestions and criticisms. I would recommend publication.

Author Response

We thank the reviewer for his constructive comments, which have certainly contributed to a significant improvement of the manuscript.

Reviewer 2 Report

The authors addressed the comments but English still needs improvements. Please use a grammar correction software or ask an English native speaker to read it.

Author Response

The authors want to thank to the Reviewer due to his/her constructive comments that allowed an important improvement of the manuscript. Following his/her comments, the manuscript was submitted to MDPI English edditing service and it was accordingly corrected. 

It can be seem in the certificate attached as "supplementary information"